# Configurations associated with the efficiency of the ophthalmology departments in public hospitals of Central South China

Yimeng Li[1], Yao Chen[1], Bosheng Ma[2], Jyu-lin Chen[3], Jie Zhong[4], Yan Jiang[1], Jing Luo[2]*, Jia Guo[1]*

1 Xiangya School of Nursing, Central South University, Changsha, Hunan, PR China, 2 Department of Ophthalmology, The Second Xiangya Hospital, Central South University, Changsha, Hunan, PR China, 3 School of Nursing, University of California, San Francisco, San Francisco, California, United States of America, 4 School of Nursing, the University of Hong Kong, Hong Kong, PR China

* guojia621@163.com (JG); luojing001@csu.edu.cn (JL)

## Abstract

### Background

Improving the efficiency of ophthalmology service is a global challenge to fight vision impairment, yet there is little concrete evidence of the current efficiency status. This study aimed to examine the efficiency of ophthalmology departments in the Hunan Province, China, and determine the associating factors of low-efficiency and high-efficiency ophthalmology departments.

### Methods

This cross-sectional study included a province-level survey of ophthalmology departments of public hospitals. All the ophthalmology departments of public hospitals in Hunan Province were invited to complete an online survey on ophthalmic competence resources. Bootstrap Data Envelopment Analysis was conducted to describe the service efficiency status of the ophthalmology departments using Maxdea (version 8.0) software. Then, we employed Fuzzy Set-Qualitative Comparative Analysis to explore the recipes of low-efficiency and high-efficiency ophthalmology departments using Fs-QCA (version 3.0) software.

### Results

One hundred and ninety-five ophthalmology departments (87 in tertiary and 108 in secondary public hospitals) completed the survey. The mean efficiency score was 0.78 for ophthalmology departments in tertiary hospitals and 0.82 for secondary hospitals. The number of ophthalmologists and equipment positively contributed to the efficiency of ophthalmology departments in tertiary and secondary hospitals. While increasing the bed capacity was not always beneficial to improving the efficiency of ophthalmology departments in secondary hospitals. For ophthalmology departments in tertiary hospitals, simply increasing the number of nurses did not universally increase efficiency unless there were enough ophthalmologists and equipment to support the nurses' work. This study also revealed 2 configurations

**Data Availability Statement:** all data are fully available without restriction, as the original data has been uploaded in the form of an Excel file, named "original-dataset" and labeled as "supporting

information", and all other data underlying the findings have been reported in the manuscript.

**Funding:** Central South University through its graduate innovative research grant (grant number: 2024ZZTS0998). The funders had no role in study design, data collection and analysis, decision to publish, or preparation of the manuscript.

**Competing interests:** The authors have declared that no competing interests exist.

**Abbreviations:** NOO, Number of ophthalmologists; NON, Number of nurses; NOB, Number of open beds for ophthalmic patients; NOE, Number of ophthalmic medical equipment; NOV, Number of outpatient and emergency visits; NOS, Number of surgical operations for inpatient; NOP, Number of inpatients; DEA, Data Envelopment Analysis; Fs-QCA, fuzzy set-Qualitative Comparative Analysi.

for ophthalmology departments in secondary hospitals and 5 configurations for those in tertiary hospitals that could guide their efficiency improvement efforts.

## Conclusions

Moderate efficiency levels in ophthalmology departments at both tertiary and secondary hospitals were found. Prioritizing the number of ophthalmologists and equipment was recommended to achieve high efficiency for ophthalmology departments in tertiary and secondary hospitals. We also proposed that blindly increasing the number of beds and nurses was meaningless, and ophthalmology departments should flex the bed capacity and number of nurses after premising having high numbers of ophthalmologists and equipment.

## Introduction

Vision impairment, including blindness, is one of the most tremendous burdens on global healthcare, with a rising prevalence, notably in low-income and middle-income countries (LMICs) [1]. In 2020, the global age-standardized prevalence was 1.21% for moderate and severe vision impairment and 0.21% for blindness [2], and LMICs have 10 times the prevalence of eye diseases than developed countries [3]. Among those patients, a large proportion (80.0% in 2020) of vision impairment is avoidable if they receive timely and appropriate treatments [4]. Both low efficiency in ophthalmology and unoptimistic coverage of ophthalmology services are confirmed to be the major reasons for this discouraging outcome [5]. However, the shortage of medical resources limits the widening of coverage of ophthalmic services [6], and limited investment in medical resources remains a big issue that is difficult to solve [7]. Thus, increasing the efficiency of the ophthalmology department is a critical challenge, especially for LMICs with limited resources.

Health service efficiency is defined as the ratio of the value of health services delivered to the resources used in producing health services [8]. It can reflect the disparity between the current and maximum possible amount of output using a given set of inputs. The efficiency of medical departments (or wards) is one of the most commonly used efficiency categories because it not only impacts the patients' medical outcomes [9] but also affects hospitals' efficiency [10]. A handful of medical departments, such as operation rooms [11], emergency departments [12], and clinical laboratories [13], reported low-efficiency status. Understanding efficiency levels and areas for improvement can improve healthcare delivery and patient outcomes.

Currently, studies on ophthalmology departments mainly focus on the coverage of ophthalmic resources or services indicating that ophthalmologists, pieces of equipment, and beds from ophthalmology departments did not meet the needs [14]. Studies revealed that less than half of hospitals provided eye health services for patients [15, 16], and rural residents are more likely to experience under-service [17] either in LMICs such as Sub-Saharan African countries or in developed countries such as the US. In addition, although LMICs account for half of the global population, it is estimated that LMICs only have 1/2 of the ophthalmic resources of developed countries [18]. In all, the inadequacy of resources or services has been revealed, yet there is little literature on the efficiency of ophthalmology departments thus is urgently indicated.

China has the largest population of eye diseases, affecting more than 59.28 million people [19]. The prevalence of moderate and severe vision impairment in China increased more

rapidly than in the other G20 countries from 1990 to 2010 [20]. However, although China has been classified as an upper-middle income country, the per-capital ophthalmology resources in China are the same as the IMICs [21], leading the current capacity of eye care services in public hospitals has failed to meet the growing demand for ophthalmology service [5]. Take cataracts as an example, the prevalence of cataracts is generally comparable among countries [22], but the Cataract Surgical Rate in China is at 2,205 surgeries performed per million individuals annually [23], lower than that of developed countries and some developing countries such as India [24]. About 43.2% of patients with Diabetic Retinopathy (DR) who were actively seeking eye care services were reported to fail to get treated [25]. Moreover, the medical resources, especially the number of open beds and medical staff, were not equipped strictly based on the number of patients [26]. This may give rise to an unjustifiable allocation of these valuable medical resources and perhaps compromise their effectiveness. Under this circumstance, examining ophthalmology departments' efficiency in China, and exploring strategies to optimize the efficiency are crucial.

There are some already known internal structural factors affecting medical departments' efficiency, such as human resources [27], the number of open beds [28], and technical resources [28]. However, the main internal structural factors affecting efficiency in various medical departments are different [16]. For example, the clinic laboratory's efficiency is associated with the number of equipment and the equipment's turnout time [13], while the emergency department's efficiency is related to the number of emergency medical professionals [12]. Given the fact that ophthalmology departments have strong specialties in diagnosis and treatment with a high dependence on instruments and equipment [29], what and how internal structural factors affect ophthalmology departments' efficiency may be unique and different from other already-known departments. Thus, studies are needed to inform the strategy development for improving efficiency.

This study aimed to examine the efficiency of ophthalmology departments in Hunan Province, China. The second objective was to determine the key factors associated with efficiency. The third objective was to investigate the configurations that guide the effectiveness of ophthalmology departments with both low and high efficiency. This study may provide evidence-based data to inform how to improve ophthalmology departments' efficiency in the LMICs, thus decreasing health disparities.

## Methods

### Study design

A cross-sectional survey was conducted online among ophthalmology departments in public hospitals in Hunan Province. Hunan Province is representative of Central South China in geography (mixed plains and mountainous areas), climate (humid subtropical), demographics (Han and multi-minorities), economy, and health policy. Hospitals in China were categorized into primary, secondary, and tertiary levels according to their medical resources and tasks. There were 701 public hospitals in Hunan Province, including 164 secondary and 95 tertiary public hospitals. Only ophthalmology departments in secondary and tertiary public hospitals were invited to participate because they dominated the provision of ophthalmology services and represented the local medical level [30].

The inclusion criteria included: 1) the ophthalmology department is in a public hospital; 2) the hospital is assessed as a secondary or tertiary level; 3) the ophthalmology department provides daily outpatient or inpatient ophthalmology service. The exclusion criteria were the department's ophthalmologists work simultaneously as otolaryngologists. Study permission

was obtained from the health commission of Hunan Province. No ethical review was required as this study only collected data on all medical resources in healthcare organizations.

## Variables and measurements

To explore the efficiency of ophthalmology services in Hunan Province, this study selected variables that were commonly used and indispensable inputs and outputs in efficiency studies [31, 32]. This study selected the number of beds, ophthalmologists (occupational ophthalmologists and occupational assistant ophthalmologists), registered nurses, and specialized equipment as the input variables because they have been proven to have a clear correlation with efficiency [31]. For the same reason, the number of outpatients and emergency visits, inpatients, and inpatient surgeries as the output variables. Then, this study identified the input allocation strategies for the high and low efficiency among ophthalmology departments. The definitions of input and output variables are illustrated in Table 1.

We employed the national Ophthalmic Competence Resource questionnaire [33] to collect data on the above variables. This questionnaire was designed by the Medical Administration of the National Health and Health Commission to measure the medical institution's essential condition of eye care capacity. It contains six parts, including 220 items on the basic setup of ophthalmology, human resources, technical and instrumental resources, eye disease diagnosis capacity, treatment ability, training, and research.

## Data collection

The questionnaire was distributed to all secondary and tertiary public hospitals in the region through the online annual performance appraisal of public hospitals conducted by the Health Commission of Hunan Province. We collected data from February 10 to March 4, 2022. At the beginning of the questionnaire, we provided detailed information about the study's objective, possible benefits and risks, and the right to decline participation. It was highlighted that this was an elective part of the annual appraisal, and the data collected by the questionnaire would not affect the results of the annual appraisal. The head of the ophthalmology department could voluntarily fill out the questionnaires following a standardized guideline and send them back via emails. Two research assistants meticulously reviewed all the collected questionnaires and conducted telephone verification for any conflicting or uncertain data.

## Data analysis

The analyses were done using SPSS (Version 28.0.0.0; Armonk, NY, United States). The mean with standard deviation (SD), maximum, and minimum values of ophthalmology resources

**Table 1. The inputs, outputs, and antecedent variables used in this study.**

| Indicator | Variable | Definition |
|---|---|---|
| Input | NOO | Number of ophthalmologists |
| | NON | Number of nurses |
| | NOB | Number of open beds for ophthalmologic patients |
| | NOE | Number of ophthalmological medical equipment |
| Output | NOV | Number of outpatient and emergency visits |
| | NOS | Number of surgical operations for inpatient |
| | NOP | Number of inpatients |
| Outcome | BSE | Bootstrap-DEA efficiency score |

were generated. Student's t-tests were used to verify statistical differences in resources, including inputs and outputs, between ophthalmology departments in tertiary and secondary hospitals. The analysis was performed at a significance level of $P<0.05$ for single-sided tests. Then, the data envelopment analysis was employed to examine the ophthalmology departments' efficiency, and the fuzzy set-qualitative comparative analysis was used to explore the factors influencing efficiency outcomes.

## Data envelopment analysis (DEA)

Data envelopment analysis (DEA) [34] was a non-parametric technique to calculate the efficiency score (range from 0 to 1) of medical units. It was widely applied to measure the relative efficiency of a set of comparable decision-making units (DMUs and ophthalmology departments in this study). The comparability of DMUs depends on whether these units possess similar levels of input and output quantities. The constant return to scale (CRS) -DEA model was one of the most extensively utilized DEA models, which assumed full proportionality between all inputs and outputs. However, the CRS-DEA model's estimation of efficiency values was vulnerable to measurement bias [35]. This bias could be corrected by applying bootstrap replications [36], which was the principle of bootstrap-DEA.

In this study, the T-test demonstrated that there were significant differences in resources between ophthalmology departments in secondary and tertiary hospitals. Therefore, to make the ophthalmology departments comparable DMUs to examine their efficiency, they were categorized into two groups based on the hospital level they belonged to and analyzed separately. We used the input-oriented CRS-DEA model because increasing ophthalmic resources may lead to efficiency gains [5]. Subsequently, using the efficiency scores calculated by the CRS-DEA model as initial efficiency samples, the bootstrap-DEA model was employed to correct the potential bias caused by CRS-DEA [37]. Two thousand times iterations were selected to operate the bootstrapping process. Maxdea (version 8.0) software was employed for the CRS-DEA and bootstrap-DEA processes.

## Fuzzy set- Qualitative Comparative Analysis (fs-QCA)

The steps of fs-QCA mainly followed two established guidelines [38, 39]. Firstly, based on the set theory, Fs-QCA could analyze the variables' degree of membership to the outcome through calibration [38]. In the calibration process, the metrics of the full members, the cross-over point, and the full non-members were set as standards for determining the corresponding variables' relatively high, medium, and low levels and efficiency scores. The degree of membership, which indicated the degrees of the variables belonging to the levels, was also calculated to categorize each variable into relatively high, medium, and low levels. In this study, ophthalmology departments with efficiency scores evaluated as relatively high are considered high efficiency, while those with efficiency scores evaluated as relatively medium or low were classified as non-high efficiency. The non-high efficiency was described as low efficiency in the following steps. Then, fs-QCA could define configurations (equal to combinations of antecedent variables) that brought about certain levels of ophthalmology departments' efficiency from several different configurations by processing the necessity test and sufficient analysis [40]. In the necessity test, we set the consistency threshold as 0.9, which was the recognized standard to ensure a necessary relationship between the configurations and the outcomes [39, 41, 42], to judge whether the variable was the necessary variable that could fully explain the resulting outcome. Necessary variables with a consistency value higher than 0.9 must be excluded from sufficient analysis. The sufficient analysis would infer the configurations that lead to different efficiency levels. We set the minimum raw consistency value to 0.80 and the Proportional Reduction in

Inconsistency (PRI) consistency values to 0.75 to ensure the sufficient relationship between the configurations and the outcomes according to the guidelines [38]. Configurations with these values and higher are coded as leading to certain levels of efficiency in ophthalmology departments. The configurations would determine key factors associated with ophthalmology departments' efficiency, and the configurations shared by the high-efficiency ophthalmology departments would unveil potential strategies for low-efficiency ophthalmology departments to reallocate resources to achieve high efficiency. In this part, we focus on the configurations that lead to high efficiency, but the configurations that lead to low efficiency were also determined according to the requirements of fs-QCA. This analysis was carried out using Fs-QCA software (version 3.0).

## Results

### Data sources

Of hospitals with ophthalmology departments, 92 tertiary and 140 secondary hospitals returned the questionnaires. Five tertiary and 32 secondary hospitals were excluded since their questionnaire had inconsistent answers (e.g., they provided ophthalmology service but did not have an ophthalmologist), and they did not provide accurate information in telephone verification. Finally, 87 tertiary and 108 secondary hospitals were involved in the analysis. The response rate of this survey was 63.39%, and the effective recovery rate was 84.05%. The sample ophthalmology departments cover all regions in Hunan Province.

### Medical resources of secondary hospitals and tertiary hospitals

The differences in resources between tertiary and secondary hospitals are illustrated in Table 2. For input variables, there were significant differences in the number of ophthalmologists, nurses, equipment, and open beds between tertiary and secondary hospitals (P<0.01), representing that the ophthalmology departments in tertiary hospitals own more inputs than the secondary hospitals. For output variables, there were significant differences in the number of inpatients, surgery, and outpatients between tertiary and secondary hospitals (P<0.01), indicating that ophthalmology departments in tertiary hospitals took care of more patients and undertook more surgeries.

### Efficiency score of ophthalmology departments in secondary hospitals and tertiary hospitals

Table 3 illustrates the CRS-DEA efficiency scores and hospitals' corresponding bootstrap-DEA efficiency scores. For secondary hospitals, the CRS-DEA efficiency scores had a mean of 0.81, a standard deviation of 0.28, a maximum value of 1.00, and a minimum value of 0.18. After the Bootstrap-DEA method, the efficiency values for secondary hospitals showed a skewed distribution with a mean of 0.82, a standard deviation of 0.27, a maximum value of 1.00, and a minimum value of 0.19.

For tertiary hospitals, the CRS-DEA efficiency scores had a mean of 0.74, a standard deviation of 0.25, a maximum value of 1.00, and a minimum value of 0.11. After the bootstrap-DEA model, the efficiency values for tertiary hospitals showed a skewed distribution with a mean of 0. 78, a standard deviation of 0.23, a maximum value of 1.00, and a minimum value of 0.15. The frequency distribution of the efficiency of the ophthalmology hospital is presented in Figs 1 and 2.

**Table 2. Comparison of the input and output variables of ophthalmology departments in secondary and tertiary hospitals.**

| Variable | Group | Mean | Max | Min | St.dev | Difference of means | t | p |
|---|---|---|---|---|---|---|---|---|
| NOB | Secondary | 22.44 | 165 | 0 | 19.82 | 10.97 | 3.31 | < .001 |
| | Tertiary | 33.41 | 200 | 0 | 26.50 | | | |
| NOO | Secondary | 5.19 | 43 | 1 | 4.53 | 6.91 | 6.44 | < .001 |
| | Tertiary | 12.09 | 59 | 2 | 9.95 | | | |
| NON | Secondary | 7.43 | 38 | 0 | 5.41 | 5.02 | 4.92 | < .001 |
| | Tertiary | 12.45 | 63 | 0 | 8.71 | | | |
| NOE | Secondary | 16.94 | 92 | 1 | 9.88 | 23.13 | 9.50 | < .001 |
| | Tertiary | 40.07 | 113 | 2 | 22.52 | | | |
| NOV | Secondary | 9494.11 | 131827 | 30 | 17104.90 | 14637.99 | 3.79 | < .001 |
| | Tertiary | 244132.10 | 241321 | 700 | 35205.73 | | | |
| NOP | Secondary | 732.00 | 7287 | 0 | 1141.574 | 923.95 | 3.80 | < .001 |
| | Tertiary | 1655.95 | 17096 | 0 | 2186.94 | | | |
| NOS | Secondary | 567.53 | 10531 | 0 | 784.23 | 1185.00 | 4.68 | < .001 |
| | Tertiary | 17552.53 | 16721 | 0 | 2482.45 | | | |

## Variables calibration for fuzzy sets

We calibrated the value of antecedent variables to a fuzzy set scale using external standards. For BSE, NOO, NOE, NOB, and NOE, their original values that covered 95%, 50%, and 5% of the data values were established as the point of full membership, crossover point, and full non-membership [43]. The calibration scores are listed in Table 4.

## Analysis of necessary conditions

For secondary hospitals, the consistency values ranged from 0.41 to 0.78. For tertiary sample hospitals, the consistency values ranged from 0.33 to 0.86. No variable had a consistency value higher than 0.9. As a result, all the variables were retained for Analysis of sufficient conditions. Table 5 represents the outcome of the test of the necessary conditions.

## Analysis of sufficient conditions

We explored the combinations of conditions that lead to high-efficiency ophthalmology departments in secondary and tertiary hospitals. The results are shown in Table 6. For secondary hospitals, two configurations that led to high-efficiency ophthalmology departments include being equipped with the high number of open beds, equipment, and ophthalmologists, although the number of nurses was low.

For tertiary hospitals, resource allocation in high-efficiency ophthalmology departments could be concluded into 5 configurations, and these five configurations could be categorized

**Table 3. CRS-DEA and Bootstrap-DEA efficiency score of ophthalmology departments in secondary and tertiary hospitals.**

| | Secondary hospital | | | | | Tertiary Hospital | | | | |
|---|---|---|---|---|---|---|---|---|---|---|
| | CRS-DEA | Bootstrap-DEA | Bias | LB | UB | CRS-DEA | Bootstrap-DEA | Bias | LB | UB |
| Mean | 0.81 | 0.82 | -0.01 | 0.71 | 0.89 | 0.74 | 0.78 | -0.03 | 0.69 | 0.86 |
| St.dev | 0.28 | 0.27 | 0.04 | 0.27 | 0.26 | 0.25 | 0.23 | 0.49 | 0.24 | 0.21 |
| Max | 1.00 | 1.00 | 0.22 | 1.00 | 1.00 | 1.00 | 1.00 | 0.12 | 1.00 | 1.00 |
| Min | 0.18 | 0.19 | -0.13 | 0.02 | 0.02 | 0.11 | 0.15 | -0.13 | 0.10 | 0.20 |

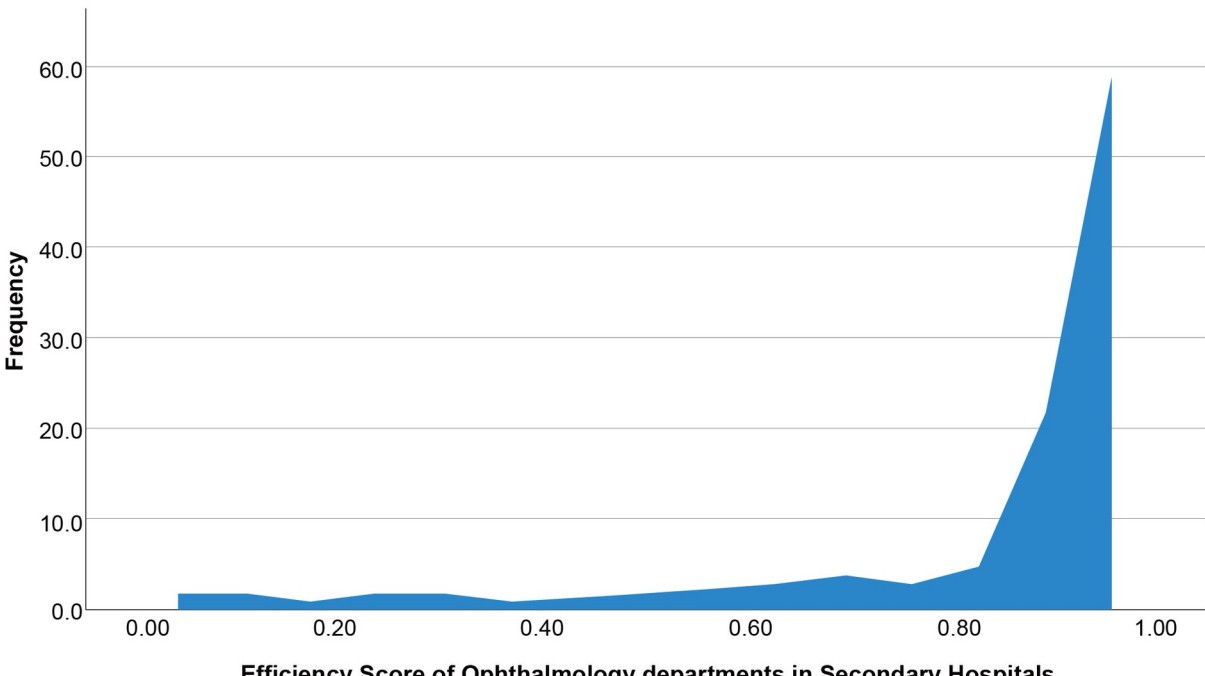

**Fig 1. Distribution of the efficiency score of ophthalmology departments in secondary hospitals.**

into 3 types. The first type included the configuration HT1 and HT2. They revealed that the combination of the high number of ophthalmologists and equipment could lead to high-efficiency ophthalmology departments in tertiary hospitals. Under these circumstances, the high-

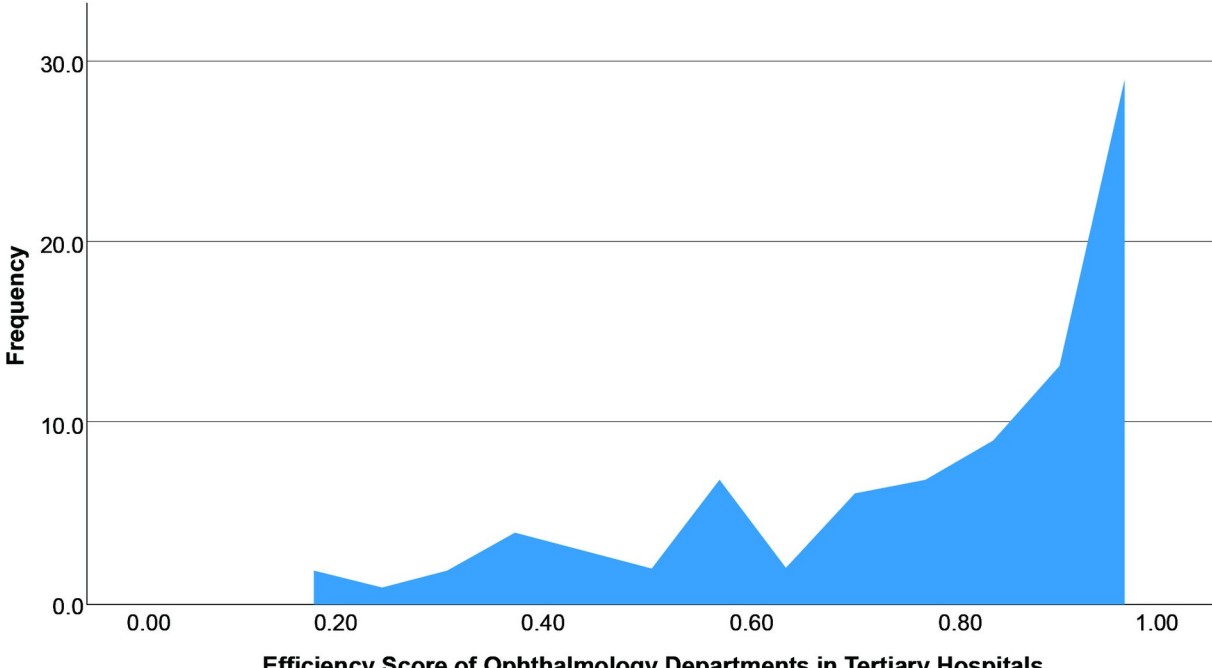

**Fig 2. Distribution of the efficiency score of ophthalmology departments in tertiary hospitals.**

**Table 4. The calibration point value of antecedent variables and outcome variable.**

| Group | Calibration point | BSE | NOO | NON | NOB | NOE |
|---|---|---|---|---|---|---|
| Secondary Hospital | Full membership | 0.99 | 10.00 | 15.00 | 48.30 | 27.00 |
| | Cross-over point | 0.95 | 4.00 | 7.00 | 20.00 | 16.00 |
| | Non-membership | 0.87 | 2.00 | 0 | 3.20 | 5.35 |
| Tertiary Hospital | Full membership | 1.00 | 28.70 | 24.00 | 70.50 | 90.70 |
| | Cross-over point | 0.86 | 9.00 | 11.00 | 30.00 | 34.00 |
| | Non-membership | 0.43 | 3.00 | 2.00 | 10.00 | 17.00 |

efficiency ophthalmology departments' quantity of open beds could be at a low level. Configuration HT3 and HT4 concluded that the second type of high-efficiency ophthalmology departments effectively combined a high number of ophthalmologists, equipment, and nurses, with a particular emphasis on the highest levels of ophthalmologists and equipment. Configuration HT5 revealed the third type of configuration that supported a high-efficiency ophthalmology department in tertiary hospitals. These high-efficiency ophthalmology departments in tertiary hospitals had high number of ophthalmologists and equipment to make up for the low levels of number of open beds and nurses.

The configurations that led to low efficiency were summarized in Table 7. The two configurations for ophthalmology departments in secondary hospitals pointed out that the combination of low number of ophthalmologists, nurses, and equipment led to low-efficiency ophthalmology departments in secondary hospitals, even though a high the number of open beds existed. For ophthalmology departments in tertiary hospitals, two types of configurations that cause low efficiency were found. The first type included the configuration LT1 and LT2. They revealed that the combination of low number of open beds, ophthalmologists, and equipment could lead to low-efficiency ophthalmology departments in tertiary hospitals.

**Table 5. Necessity of the conditions relative to the high and low efficiency.**

| | | High efficiency | | Low efficiency | |
|---|---|---|---|---|---|
| | | Consistency | Coverage | Consistency | Coverage |
| NOO | Secondary | 0.71 | 0.63 | 0.51 | 0.45 |
| | Tertiary | 0.78 | 0.86 | 0.45 | 0.45 |
| ~NOO | Secondary | 0.41 | 0.44 | 0.60 | 0.68 |
| | Tertiary | 0.50 | 0.50 | 0.86 | 0.78 |
| NON | Secondary | 0.58 | 0.58 | 0.50 | 0.53 |
| | Tertiary | 0.75 | 0.78 | 0.54 | 0.51 |
| ~NON | Secondary | 0.53 | 0.50 | 0.60 | 0.60 |
| | Tertiary | 0.53 | 0.56 | 0.77 | 0.73 |
| NOB | Secondary | 0.57 | 0.56 | 0.54 | 0.56 |
| | Tertiary | 0.67 | 0.80 | 0.45 | 0.48 |
| ~NOB | Secondary | 0.56 | 0.54 | 0.78 | 0.59 |
| | Tertiary | 0.57 | 0.53 | 0.82 | 0.69 |
| NOE | Secondary | 0.65 | 0.65 | 0.44 | 0.46 |
| | Tertiary | 0.75 | 0.82 | 0.48 | 0.48 |
| ~NOE | Secondary | 0.46 | 0.44 | 0.67 | 0.67 |
| | Tertiary | 0.52 | 0.53 | 0.82 | 0.75 |

[a] ~ indicates the negation of a condition.

**Table 6. Configuration of conditions for high-efficiency ophthalmology departments in secondary and hospitals.**

|  |  | NOB | NOO | NON | NOE | Raw Consistency | Raw Coverage | Unique Coverage | Overall Consistency | Overall Coverage |
|---|---|---|---|---|---|---|---|---|---|---|
| Secondary hospitals | Configuration HS1 | ● | ● | ⊗ | ● | 0.81 | 0.20 | 0.20 | 0.81 | 0.20 |
|  | Configuration HS2 | ● | ● | ⊗ | ● | 0.81 | 0.20 | 0.20 |  |  |
| Tertiary hospitals | Configuration HT1 | ⊗ | ● |  | ● | 0.89 | 0.42 | 0.04 | 0.85 | 0.71 |
|  | Configuration HT2 | ⊗ | ⬤ |  | ● | 0.89 | 0.42 | 0.04 |  |  |
|  | Configuration HT3 |  | ● | ● | ● | 0.92 | 0.65 | 0.24 |  |  |
|  | Configuration HT4 |  | ⬤ | ● | ● | 0.92 | 0.65 | 0.24 |  |  |
|  | Configuration HT5 | ● | ⊗ | ● | ⊗ | 0.82 | 0.30 | 0.02 |  |  |

[a]HS means high efficiency in ophthalmology departments in secondary hospitals, and HT means high efficiency in ophthalmology departments in tertiary hospitals.
[b]Full black circles and crossed-out circles indicate the presence and the absence of causal conditions, respectively. Large circles indicate the core conditions, small circles indicate the peripheral conditions and the blank cells represent conditions that do not matter for the solution.

Configuration LT3 and LT4 concluded the configurations of the second type of low-efficiency ophthalmology departments in tertiary hospitals: they had a combination of high number of nurses and open beds, and low number of ophthalmologists and equipment. Among these four configurations that led to low-efficiency ophthalmology departments in tertiary hospitals, the lowest levels of the number of ophthalmologists and equipment were highlighted.

The configurations described above disclose potential reasons for high-efficiency and low-efficiency ophthalmology departments in both secondary and tertiary hospitals. All these configurations had consistency values of more than 0.8, indicating that the relationships between the configurations and the outcome (low or high-efficiency score) were credible.

## Robustness test

Changing the consistency level was the guideline-recommended and most commonly used method to test the robustness of the fs-QCA results [39, 44]. This study followed the method,

**Table 7. Configurations for low-efficiency ophthalmology departments in secondary and hospitals.**

|  |  | NOB | NOO | NON | NOE | Raw Consistency | Raw Coverage | Unique Coverage | Overall Consistency | Overall Coverage |
|---|---|---|---|---|---|---|---|---|---|---|
| Secondary hospitals | Configuration LS1 | ● | ⊗ | ⊗ | ⊗ | 0.85 | 0.13 | 0.13 | 0.85 | 0.13 |
|  | Configuration LS2 | ● | ⊗ | ⊗ | ⊗ | 0.85 | 0.13 | 0.13 |  |  |
| Tertiary hospitals | Configuration LT1 | ⊗ | ⊗ |  | ⊗ | 0.80 | 0.70 | 0.26 | 0.80 | 0.75 |
|  | Configuration LT2 | ⊗ | ⊗ |  | ⊗ | 0.80 | 0.70 | 0.26 |  |  |
|  | Configuration LT3 |  | ⊗ | ● | ⊗ | 0.83 | 0.48 | 0.05 |  |  |
|  | Configuration LT4 |  | ⊗ | ● | ⊗ | 0.83 | 0.48 | 0.05 |  |  |

[a]HS means high efficiency in ophthalmology departments in secondary hospitals, and HT means high efficiency in ophthalmology departments in tertiary hospitals.
[b]Full black circles and crossed-out circles indicate the presence and the absence of causal conditions, respectively. Large circles indicate the core conditions, small circles indicate the peripheral conditions and the blank cells represent conditions that do not matter for the solution.

**Table 8. Robustness test result with the consistency level of 0.70.**

| | | NOB | NOO | NON | NOE | Raw Consistency | Raw Coverage | Unique Coverage | Overall Consistency | Overall Coverage |
|---|---|---|---|---|---|---|---|---|---|---|
| Secondary hospitals | Configuration HS1 | ● | ● | ⊗ | ● | 0.81 | 0.20 | 0.20 | 0.81 | 0.20 |
| | Configuration HS2 | ● | ● | ⊗ | ● | 0.81 | 0.20 | 0.20 | | |
| Tertiary hospitals | Configuration HT1 | ⊗ | ● | | ● | 0.89 | 0.42 | 0.04 | 0.78 | 0.73 |
| | Configuration HT2 | ⊗ | ● | | ● | 0.89 | 0.42 | 0.04 | | |
| | Configuration HT3 | | ● | ● | ● | 0.92 | 0.65 | 0.24 | | |
| | Configuration HT4 | | ● | ● | ● | 0.92 | 0.65 | 0.24 | | |
| | Configuration HT5 | ● | ⊗ | ● | ⊗ | 0.70 | 0.37 | 0.04 | | |

[a]HS means high efficiency in ophthalmology departments in secondary hospitals, and HT means high efficiency in ophthalmology departments in tertiary hospitals.
[b]Full black circles and crossed-out circles indicate the presence and the absence of causal conditions, respectively. Large circles indicate the core conditions, small circles indicate the peripheral conditions and the blank cells represent conditions that do not matter for the solution.

changing the raw consistency level from 0.80 to 0.90 and 0.70 to conduct the robustness test. When the consistency level was 0.70, the solution results are in Table 8, and when the consistency level was 0.90, the solution results are in Table 9. The results had no substantial changes in the main findings, except for minor changes in the value of coverage and consistency, which indicated the relative robustness of the results of this study.

## Discussion

This study first examined the ophthalmology departments from an efficiency perspective and revealed that ophthalmology departments in both tertiary and secondary hospitals in Hunan Province of Central South China had underutilized their ophthalmology resources. This result demonstrated that besides increasing inputs and costs as prior studies suggested [45],

**Table 9. Robustness test result with the consistency level of 0.90.**

| | | NOB | NOO | NON | NOE | Raw Consistency | Raw Coverage | Unique Coverage | Overall Consistency | Overall Coverage |
|---|---|---|---|---|---|---|---|---|---|---|
| Secondary hospitals | Configuration HS1 | ● | ● | ⊗ | ● | 0.83 | 0.30 | 0.22 | 0.81 | 0.20 |
| | Configuration HS2 | ● | ● | ⊗ | ● | 0.83 | 0.30 | 0.22 | | |
| Tertiary hospitals | Configuration HT1 | ⊗ | ● | | ● | 0.89 | 0.42 | 0.04 | 0.85 | 0.71 |
| | Configuration HT2 | ⊗ | ● | | ● | 0.89 | 0.42 | 0.04 | | |
| | Configuration HT3 | | ● | ● | ● | 0.92 | 0.65 | 0.24 | | |
| | Configuration HT4 | | ● | ● | ● | 0.92 | 0.65 | 0.24 | | |
| | Configuration HT5 | ● | ⊗ | ● | ⊗ | 0.70 | 0.37 | 0.04 | | |

[a]HS means high efficiency in ophthalmology departments in secondary hospitals, and HT means high efficiency in ophthalmology departments in tertiary hospitals.
[b]Full black circles and crossed-out circles indicate the presence and the absence of causal conditions, respectively. Large circles indicate the core conditions, small circles indicate the peripheral conditions and the blank cells represent conditions that do not matter for the solution.

optimizing resource utilization was also an approach to improve the efficiency of ophthalmology departments. In our study, configurations that led to high efficiency were identified. High efficiency in ophthalmology departments was consistently associated with more ophthalmologists and equipment in both secondary and tertiary hospitals. Simultaneously, in ophthalmology departments of secondary hospitals, maintaining a high number of open beds is crucial, while ophthalmology departments in tertiary hospitals emphasize the urgent necessity for an increase of nurses. The differences in configurations of high-efficiency ophthalmology departments between secondary and tertiary hospitals disclosed that the efficiency of ophthalmology departments in secondary and tertiary hospitals should be increased by following different resource allocation strategies.

The efficiency scores revealed by this study indicated that these ophthalmology departments could have increased efficiency by 18% in secondary hospitals and 22% in tertiary hospitals with the same amount of input. The potential for enhancing efficiency primarily stems from the underutilization of available resources in ophthalmology departments [46]. Thus, our study provides evidence of the positive impact on efficiency by augmenting the overall resource utilization rate. This is consistent with previous studies that ensuring resource utilization efficiency is a necessary condition for improving medical efficiency [47]; otherwise, increasing medical resources may not necessarily be conducive to the enhancement of medical services [48].

This study concluded configurations that contribute to the high efficiency of ophthalmology departments. For ophthalmology departments in secondary hospitals, we found that high efficiency was associated with more open beds, equipment, and ophthalmologists coupled with a lower number of nurses. The results challenged previous studies regarding augmenting the number of nurses can not only improve health outcomes for patients [49] but also benefit enhancing medical efficiency [50]. There may be two reasons. First, the marginal adverse impact of a lower number of nurses on the efficiency of ophthalmology departments may be attributed to the relatively low demand for nursing care from patients in secondary hospitals [51]. Due to the National Health and Health Commission's requirements for ophthalmology service development [52] and the patient's preference for the tertiary hospitals in China [53], ophthalmology departments in secondary hospitals mainly provide services to fewer patients with less severe diseases requiring less or simple care from nurses [54]. In addition, inconsistencies in the results can be ascribed to the distinctive characteristics of ophthalmology departments compared to the other medical departments reported in previous studies where the specific ophthalmology diagnosis and treatment heavily rely on the number of ophthalmologists and equipment [29]. Therefore, unlike nurses who could fulfill their care responsibilities independently [49], the effectiveness of nurses' engagement in ophthalmology services relied on the availability of adequate numbers of ophthalmologists and equipment [55], with the adequate number of equipment and ophthalmologists reducing the nurse workforce [56]. Therefore, the relatively low workload could be dealt with a low number of nurses [57], thereby not reducing efficiency.

For highly efficient ophthalmology departments in tertiary hospitals, besides the higher levels of a number of equipment and ophthalmologists, they are typically equipped with a higher level number of nurses or a lower level number of open beds. Surprisingly, a low number of open beds was not negatively associated with high efficiency supported by high numbers of ophthalmologists and equipment. This is consistent with some previous studies indicating that an increase in the number of beds does not necessarily result in improved efficiency [50]. This may be because the high number of ophthalmologists and equipment were related to the higher quality of treatments, which could reduce the length of stay [58] and increase the bed utilization rate. This may also be because having a large number of beds not only allowed for

the possibility of accommodating more patients, it also brought the risk of bed wastage, and therefore the utilization of bed resources was more crucial than the number of beds [59].

In tertiary hospitals, a small subset of highly efficient ophthalmology departments was found to be characterized by more nurses and open beds alongside less equipment and ophthalmologists. This may be explained by the cooperation among ophthalmology departments in different hospitals [60]. Some ophthalmology departments only focused on information collection and basic ophthalmology care, with support from a high number nursing staff and available beds [61]. Meanwhile, ophthalmologists and equipment in other hospitals were tasked with diagnosing and treating patients through telemedicine or referral [62].

## Limitations

First, considering the relevance of variables to the research objectives and the findings from the literature, this study only selected variables that theoretically most influenced efficiency (having been proven to have a clear correlation with efficiency). Consequently, some factors that could affect efficiency were not included, such as indicators representing the quality of care, patient outcomes, and financial variables. In addition, the revealed efficiency scores only described the efficiency level of ophthalmology departments in public hospitals in Hunan Province, thus could not be generalized to the efficiency of other developing countries. Finally, a larger sample size is necessary for a more comprehensive understanding of ophthalmology efficiency across broader regions.

## Implications

There are clinical implications for this study. For all ophthalmology departments, prioritizing the number of ophthalmologists and equipment simultaneously is crucial for improving efficiency. Our findings also suggest that when reducing open beds to cut medical costs [28], ophthalmology departments should maintain adequate numbers of equipment and ophthalmologists to ensure the number of patients they can serve is not reduced due to fewer beds. In addition, for low-efficiency ophthalmology departments, the configurations that led to the high efficiency of ophthalmology departments could serve as effective strategies for efficiency improvement in China.

We also provided the following insights and guidance for future research. First, the efficiency scores found in the current study implied that the existing ophthalmology resources were not fully utilized. Studies were needed to help reduce resource waste and further increase efficiency. Also, the efficiency configurations identified in the study for ophthalmology departments provided evidence of the diverse functions of factors influencing efficiency. Additional case studies were recommended to elucidate the scientific explanations behind these configurations.

## Conclusion

Appropriate resource utilization could increase the efficiency of ophthalmology departments. The key factors affecting efficiency were the number of ophthalmologists and equipment; jointly increasing them could benefit the ophthalmology department's efficiency more than increasing them separately. Tailored configurations were provided to guide efficiency improvement in ophthalmology departments among different levels of hospitals of some LMICs, thus somewhat reducing health disparities.

## Supporting information

**S1 File. Original-dataset.**
(XLSX)

## Acknowledgments

The authors thank Dr. Xiao Liu for her support and valuable input to recruitment and data collection.

## Author Contributions

**Conceptualization:** Jing Luo, Jia Guo.

**Data curation:** Bosheng Ma.

**Formal analysis:** Yimeng Li, Bosheng Ma.

**Funding acquisition:** Yimeng Li.

**Supervision:** Jia Guo.

**Writing – original draft:** Yimeng Li.

**Writing – review & editing:** Yimeng Li, Yao Chen, Jyu-lin Chen, Jie Zhong, Yan Jiang, Jing Luo, Jia Guo.

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
