## [Decision Letter · Decision Letter 0]

13 Nov 2024

PONE-D-24-25670Configurations associated with the efficiency of the ophthalmology departments in public hospitals of Central South ChinaPLOS ONE

Dear Dr. Li,

Thank you for submitting your manuscript to PLOS ONE. After careful consideration, we feel that it has merit but does not fully meet PLOS ONE’s publication criteria as it currently stands. Therefore, we invite you to submit a revised version of the manuscript that addresses the points raised during the review process.

We look forward to receiving your revised manuscript.

Kind regards,

Andrzej Grzybowski

Academic Editor

PLOS ONE

Journal Requirements:

2. Thank you for stating the following financial disclosure: [Central South University through its graduate innovative research grant (grant number: 2024ZZTS0998)].

3. In the online submission form, you indicated that [Data cannot be shared publicly because other studies using this data have not been fully completed. Data are available from the corresponding author upon reasonable request].

Reviewers' comments:

Reviewer's Responses to Questions

**Comments to the Author**

1. Is the manuscript technically sound, and do the data support the conclusions?

Reviewer #1: Yes

Reviewer #2: Yes

2. Has the statistical analysis been performed appropriately and rigorously? 

Reviewer #1: Yes

Reviewer #2: Yes

3. Have the authors made all data underlying the findings in their manuscript fully available?

Reviewer #1: Yes

Reviewer #2: Yes

4. Is the manuscript presented in an intelligible fashion and written in standard English?

Reviewer #1: Yes

Reviewer #2: Yes

5. Review Comments to the Author

Reviewer #1: The paper addresses a relevant topic and aligns with the requirements and focus of the journal. What I appreciate the most is:

Unique use of combined methods: The study combines Data Envelopment Analysis (DEA) and Fuzzy Set-Qualitative Comparative Analysis (fs-QCA), which is a relatively innovative approach to measuring the efficiency of healthcare facilities. DEA provides precise quantitative data on efficiency, while fs-QCA identifies specific configurations of variables that lead to high or low efficiency. This approach not only evaluates efficiency but also uncovers specific combinations of factors influencing it, which is uncommon in typical efficiency research.

Overall, this research provides a new perspective on efficiency in ophthalmology, an innovative methodological approach, and practical recommendations that could be useful not only for the Chinese healthcare system but also for other developing countries facing similar challenges.

Reviewer #2: Overall, I thought your article was well-written and provided sound analysis of the determinants of efficiency in ophthalmology departments. I appreciated your discussion of China's context for ophthalmological health as well as the policy implications of your fundings. As PLOS ONE is an interdisciplinary journal, some additional explanations of what is considered a "high", "low", or "adequate" number of resources/staff that you are measuring here would be helpful in assessing the impact of your results. I have provided some additional minor comments below.

Line 23 - China is now classified as an upper-middle income country, not sure if it’s appropriate to call “a typical LMIC.”

Line 242 - What does degree of membership to the outcome mean? This section could use more explanatory text.

Line 253 - Why was the threshold 0.9? Is this standard practice in the literature?

Lines 258-259 - Same as above, why were these values selected?

Lines 338-341 - This sentence is confusingly worded. Suggest rephrasing “high levels of the number” to be “high numbers”; combine lines 340 & 341 into one sentence.

Lines 343-345 - Same as above, change “high levels of the number” to “high numbers”. Please change in subsequent lines as well.

Lines 380-384 - Please expand this section - did you perform a sensitivity analysis? The methods used to check robustness are not clear. A table would be helpful here as well.

Lines 389-390 - Please change wording of “didn’t fully utilize their ophthalmology resources” to “had underutilized their ophthalmology resources”.

Line 438 - Again, lower number of nurses or a lower number of open beds.

Line 452 - Citation needed.

Lines 472-473 - What is considered an adequate number?

6. PLOS authors have the option to publish the peer review history of their article (what does this mean?). If published, this will include your full peer review and any attached files.

Reviewer #1: No

Reviewer #2: No

---

## [Author Response · Author response to Decision Letter 0]

20 Nov 2024

Dear Editors and Reviewers:

Thank you for your letter and for the reviewers’ comments concerning our manuscript entitled “Configurations associated with the Efficiency of the Ophthalmology Departments in Public Hospitals of Central South China (PONE-D-24-25670)”. Those comments are all valuable and very helpful for revising and improving our paper. We have studied the comments carefully, and according to these comments and suggestions, we added new information to the revised manuscript to help clarify the content. The response to each comment and corresponding changes are listed as follows.

① The review’s comment: China is now classified as an upper-middle income country, not sure if it’s appropriate to call “a typical LMIC.”

The authors’ answer: Thank you very much for your suggestions. We have reviewed the literature again and confirmed that, although China is currently an upper-middle-income country, its per capita ophthalmic resources are comparable to those of typical LMICs and fall short compared to other developed countries. Therefore, we revised the sentence to highlight the current state of ophthalmic resources in China, ensuring the text is more rigorous.

Revised sentence: China has the largest population of eye diseases, affecting more than 59.28 million people [19]. The prevalence of moderate and severe vision impairment in China increased more rapidly than in the other G20 countries from 1990 to 2010[20]. However, although China has been classified as an upper-middle income country, the per-capital ophthalmology resources in China are the same as the LMICs [21], leading the current capacity of eye care services in public hospitals has failed to meet the growing demand for ophthalmology service [5]. (revised manuscript: lines 98-104)

② The review’s comment: What does degree of membership to the outcome mean? This section could use more explanatory text.

The authors’ answer: Thank you for your valuable feedback. The degree of membership represents the extent to which a variable belongs to a specific outcome set. We have added an explanation of "the degree of membership" in line 247.

Revised sentence: In the calibration process, the metrics of the full members, the cross-over point, and the full non-members were set as standards for determining the corresponding variables' relatively high, medium, and low levels and efficiency scores. The degree of membership, which indicated the degrees of the variables belonging to the levels, was also calculated to categorize each variable into relatively high, medium, and low levels. (revised manuscript: lines 219-224)

③ The review’s comment: Why was the threshold 0.9? Is this standard practice in the literature? Same as above, why were these values selected?

The authors’ answer: Thank you for pointing out these problems. We have added the rationale and relevant citations for selecting these values in lines 257 and 264–265.

Revised sentence: 

In the necessity test, we set the consistency threshold as 0.9, which was the recognized standard to ensure a consistent relationship between the configurations and the outcomes [39,41,42], to judge whether the variable was the necessary variable that could fully explain the resulting outcome. (revised manuscript: lines 256-260)

The sufficient analysis would infer the configurations that lead to different efficiency levels. We set the minimum raw consistency value to 0.80 and the Proportional Reduction in Inconsistency (PRI) consistency values to 0.75 to ensure the sufficient relationship between the configurations and the outcomes according to the guidelines [38]. (revised manuscript: lines 231-240)

④ The review’s comment: This sentence is confusingly worded. Suggest rephrasing “high levels of the number” to be “high numbers”; combine lines 340 & 341 into one sentence. 

The authors’ answer: Thank you for your thorough review. Following your suggestion, we have revised "high levels of the number" to "high numbers." The sentence consolidation has also been revised accordingly, and the updated sentence can be found in lines 344–347.

Revised sentence: 

Configuration HT3 and HT4 concluded that the second type of high-efficiency ophthalmology departments effectively combined a high number of ophthalmologists, equipment, and nurses, with a particular emphasis on the highest levels of ophthalmologists and equipment (revised manuscript: lines 320–323). 

⑤ The review’s comment: Please change wording of “didn’t fully utilize their ophthalmology resources” to “had underutilized their ophthalmology resources”. 

The author's answer: Thank you for your suggestion. We have revised the sentence accordingly.

Revised sentence: 

This study first examined the ophthalmology departments from an efficiency perspective and revealed that ophthalmology departments in both tertiary and secondary hospitals in Hunan Province of Central South China had underutilized their ophthalmology resources. (revised manuscript: lines 385-388). 

⑥ The review’s comment: Again, lower number of nurses or a lower number of open beds.

The authors’ answer: Thank you for your thorough review. Following your suggestion, we have revised "low levels of the number" to "low numbers." 

⑦ The review’s comment: Please expand this section - did you perform a sensitivity analysis? The methods used to check robustness are not clear. A table would be helpful here as well.

The authors’ answer: Thank you for your valuable feedback. Based on your suggestions, we have rewritten this section, adding relevant content to clarify this part more effectively. Additionally, we have included the table with the robustness test results in the manuscript.

Revised sentence: 

Changing the consistency level was the guideline-recommended and most commonly used method to test the robustness of the fs-QCA results [39,43]. This study followed the method, changing the raw consistency level from 0.80 to 0.90 and 0.70 to conduct the robustness test. When the consistency level was 0.70, the solution results are in Table 8, and when the consistency level was 0.90, the solution results are in Table 9. The results had no substantial changes in the main findings, except for minor changes in the value of coverage and consistency, which indicated the relative robustness of the results of this study. (revised manuscript: lines 363-371).

⑧ The review’s comment: Citation needed.

The authors’ answer: Thank you for your feedback. We have carefully reviewed the literature and added supporting references in the relevant part of this sentence.

Revised sentence: 

This may be explained by the cooperation among ophthalmology departments in different hospitals [59]. (revised manuscript: lines 448-449)

⑨ The review’s comment: What is considered an adequate number?

The authors’ answer: Thank you for your suggestion. We have rewritten this sentence to clarify that "adequate" refers to ensuring that the number of facilities and ophthalmologists can maintain the hospital's capacity for ophthalmic patients, even with a reduction in the number of beds. The specific revision can be found in lines 494–495.

Revised sentence: 

Our findings also suggest that when reducing open beds to cut medical costs [28], ophthalmology departments should maintain adequate numbers of equipment and ophthalmologists to ensure the number of patients they can serve is not reduced due to fewer beds. (revised manuscript: lines 468-471).

In addition to the above revisions, we have also updated the reference list by adding the following citations (as numbered in the revised manuscript). Apart from this, we have not removed or made any other changes to the references.

[21] Resnikoff S, Lansingh VC, Washburn L, Felch W, Gauthier T-M, Taylor HR, et al. Estimated number of ophthalmologists worldwide (International Council of Ophthalmology update): will we meet the needs? Br J Ophthalmol. 2020;104: 588–592. doi:10.1136/bjophthalmol-2019-314336

[39] Schneider CQ, Wagemann C. Set-Theoretic Methods for the Social Sciences: A Guide to Qualitative Comparative Analysis. Cambridge: Cambridge University Press; 2012. doi:10.1017/CBO9781139004244

[41] Fiss PC. Building better causal theories: A fuzzy set approach to typologies in organizational research. Acad Manage J. 2011;54: 393–420. doi:10.5465/AMJ.2011.60263120

[42] Gupta K, Crilly D, Greckhamer T. Stakeholder engagement strategies, national institutions, and firm performance: A configurational perspective. Strateg Manag J. 2020;41: 1869–1900. doi:10.1002/smj.3204

[43] White L, Lockett A, Currie G, Hayton J. Hybrid Context, Management Practices and Organizational Performance: A Configurational Approach. J Manag Stud. 2021;58: 718–748. doi:10.1111/joms.12609

[59] Wang J, Ju R, Chen Y, Zhang L, Hu J, Wu Y, et al. Automated retinopathy of prematurity screening using deep neural networks. EBioMedicine. 2018;35: 361–368. doi:10.1016/j.ebiom.2018.08.033

We tried our best to improve the manuscript and made some changes marked in red in the revised paper, which will not influence the content and framework of the paper. We appreciate your warm work earnestly and hope the correction will meet with approval. Once again, thank you very much for your comments and suggestions.

Sincerely,

Jia Guo, PhD, RN, Professor (Corresponding author)

Xiangya School of Nursing, Central South University,

172 Tongzipo Road, Changsha, Hunan, PR China; 410013

E-mail: guojia621@163.com

---

## [Editor Report · Decision Letter 1]

22 Nov 2024

Configurations associated with the efficiency of the ophthalmology departments in public hospitals of Central South China

PONE-D-24-25670R1

Dear Dr. Li,

We’re pleased to inform you that your manuscript has been judged scientifically suitable for publication and will be formally accepted for publication once it meets all outstanding technical requirements.

Kind regards,

Andrzej Grzybowski

Academic Editor

PLOS ONE
---

## [Editor Report · Acceptance letter]

27 Nov 2024

PONE-D-24-25670R1 

PLOS ONE

Dear Dr. Li, 

I'm pleased to inform you that your manuscript has been deemed suitable for publication in PLOS ONE. Congratulations! Your manuscript is now being handed over to our production team.

Kind regards, 

on behalf of

Dr. Andrzej Grzybowski 

Academic Editor

PLOS ONE